# Age- and Sex-Specific Physical Fitness Reference and Association with Body Mass Index in Hong Kong Chinese Schoolchildren

**DOI:** 10.3390/ijerph192215346

**Published:** 2022-11-20

**Authors:** Ka-Man Yip, Sam W. S. Wong, Gilbert T. Chua, Hung-Kwan So, Frederick K. Ho, Rosa S. Wong, Keith T. S. Tung, Elaine Y. N. Chan, Winnie W. Y. Tso, Bik-Chu. Chow, Genevieve P. G. Fung, Wilfred H. S. Wong, Patrick Ip

**Affiliations:** 1Department of Paediatrics and Adolescent Medicine, The University of Hong Kong, Hong Kong SAR, China; 2Physical Fitness Association of Hong Kong, Hong Kong SAR, China; 3Institute of Health and Wellbeing, University of Glasgow, Glasgow G12 8QQ, UK; 4Department of Sport, Physical Education and Health, Hong Kong Baptist University, Hong Kong SAR, China; 5Department of Paediatrics, The Chinese University of Hong Kong, Hong Kong SAR, China

**Keywords:** physical fitness, reference values, BMI, Chinese children, exercise, Hong Kong

## Abstract

There is lacking a population-based study on the fitness level of Hong Kong schoolchildren, and it seems that increasing childhood obesity prevalence has shifted the classification of healthy fitness, with ‘underfit’ as normal. This cross-sectional territory study aimed to develop an age- and sex-specific physical fitness reference using a representative sample of children aged 6–17 and to determine the associations with body mass index in schoolchildren. The study analyzed Hong Kong School Physical Fitness Award Scheme data covering grade 1 to grade 12 students’ physical fitness and anthropometric measurements from 2017 to 2018. This reference was established without the impact due to COVID-19. Four aspects of physical fitness tests were measured using a standardized protocol, including (i) upper limb muscle strength, (ii) one-minute sit-up, (iii) sit-and-reach, and (iv) endurance run tests. The generalized additive model for location, scale, and shape was used to construct the reference charts. A Mann–Whitney U test was used to compare the mean differences in age, weight, and height, and a Pearson’s chi-square test was used to examine the distributions of sex groups. A Kruskal–Wallis test was used to compare the group differences in BMI status, followed by the Dunn test for pairwise comparisons. A 5% level of significance was regarded as statistically significant. Data of 119,693 students before the COVID-19 pandemic were included in the analysis. The association between physical fitness level and BMI status varied depending on the test used, and there were significant differences in fitness test scores among BMI groups. The mean test scores of the obese group were lower in most of the tests for both boys and girls, except for handgrip strength. The underweight group outperformed the obese group in push-ups, one-minute sit-ups, and endurance run tests, but not in handgrip strength. In conclusion, a sex- and age-specific physical fitness reference value for Hong Kong Chinese children aged 6 to 17 years old is established, and this study demonstrated a nonlinear relationship between BMI status and physical fitness. The reference will help to identify children with poor physical fitness to offer support and guidance on exercise training. It also serves as a baseline for assessing the impact of the COVID-19 pandemic on Hong Kong students’ physical fitness.

## 1. Introduction

Childhood physical fitness tracks moderately well into adulthood [1,2]. Physical fitness is defined as the body’s condition resulting from a lifestyle that includes a balanced cardiopulmonary function, muscle strength, muscle endurance and flexibility, and maintaining an ideal body weight [3]. According to the global non-communicable diseases action plan 2013–2020 by the World Health Organization, physical fitness is considered a public health priority [4]. The monitoring of physical fitness should be in parallel with promoting an active lifestyle among schoolchildren [5], as it gives them a sense of direction and helps them make the necessary adjustments to their daily activities. Physical fitness is an integrated measure of the body’s ability to perform physical activity that includes several parameters, such as cardiorespiratory fitness and muscle strength, all of which are important indicators of child health.

Secular declines in physical fitness coincide temporally with increases in body mass index (BMI), also observed in China during 1985–2014 [6]. The childhood obesity epidemic [7] was found to be associated with low levels of physical activity [8], which favours impaired physical fitness [9,10,11,12]. Several studies have found links between physical fitness and BMI status in Chinese children [13,14,15,16]. At the other end of the spectrum, a low BMI may have a negative impact on some measures of physical fitness [13,17]. There is a potential nonlinear relationship between them, and underweight adolescents outperformed their overweight/obese peers [9,14]. However, no population-based study on the association between physical fitness and BMI status was conducted in Hong Kong Chinese children aged 6–17. Child BMI status is influenced by nutrition, physical activity, and the environment. Urbanization frequently fosters an obesogenic environment, increasing children’s risk of developing obesity [18]. Hong Kong, as one of the most urbanized cities in China, also faces an obesity epidemic in children [19]. In addition, the activity patterns affected by the government’s social distance requirements during the COVID-19 pandemic have had a dramatic impact on physical activity and fitness in the pediatric population [20,21]. Indeed, the deconditioning effect of the COVID-19 pandemic has resulted in significant, measurable declines in cardiorespiratory fitness in healthy children [22].

Therefore, the normative values of physical fitness are essential references for monitoring the trend of physical fitness over time. Meanwhile, the School Physical Fitness Award Scheme (SPFAS) established an electronic platform to collect anthropometric and fitness data from participating students, including approximately half of all local schools in Hong Kong. It offers a good opportunity to establish reference values for a new set of fitness tests using common and well-standardized methods in a representative sample of Hong Kong schoolchildren. The present study is the first population-based study of fitness in Hong Kong children and aimed (1) to establish the age- and sex-specific fitness references using a representative sample of children aged 6–17 and (2) to determine the associations with BMI.

## 2. Materials and Methods

### 2.1. Database

This cross-sectional study was performed using existing anonymous records from the SPFAS. SPFAS was implemented in 1990 and is a territory-wide programme jointly organized by the Education Bureau, the Physical Fitness Association of Hong Kong, China and the Hong Kong Childhealth Foundation [23]. SPFAS collected detailed physical fitness data from all participating school children in Hong Kong annually and in a longitudinal fashion. It aims to assess, monitor, and promote awareness of health-related fitness and regular physical exercise. Forty-eight percent of all local schools in Hong Kong have participated in SPFAS [23]. Key aspects of students’ physical fitness and growth parameters were directly assessed using the standard protocol at the beginning of each school year. The physical fitness data was recorded with an electronic platform, allowing policymakers and participating schools to analyze and monitor their students’ fitness performance. The current study included the SPFAS data covering grade 1 through grade 12 students’ fitness and anthropometric measurements from 2017 to 2018. This study was approved by the Institutional Review Board of the University of Hong Kong/Hospital Authority Hong Kong West Cluster, and written informed consent was obtained from all participating schools (Registration number: UW18-448). Only Chinese students aged between 6 and 17 were included. Students without anthropometric data or complete data of the fitness tests were excluded.

#### 2.1.1. Anthropometric Measurements

Students’ body height and weight were measured barefoot and in lightweight clothing by trained teachers following a standardized protocol published earlier [9,23]. The body height and weight were rounded up to the nearest 0.1 cm and 100 g, respectively.

#### 2.1.2. Physical Fitness Assessment

Students were briefed on the testing procedures and practiced the procedures in another physical education class one week before the assessment to enhance their understanding. On the assessment day, students were given a 10 to 15 min warm-up session before the fitness tests. Standard demonstrations and verbal cues were provided to the students to optimize their performance during the actual tests.

Four fitness tests were carried out during the physical education classes: (i) upper limb muscle strength, (ii) one-minute sit-up test, (iii) sit-and-reach test, and (iv) endurance run tests. Each class had approximately 30 students divided into groups of 4–6 students to complete each test. All physical education teachers were provided with a detailed manual describing the standardized procedure, demonstrations to the students, and verbal cues to minimize inter-rater variability. Although no information on test–retest reliability was collected during this study, these physical fitness tests have demonstrated good concurrent validity in predicting various health outcomes and are regarded as important health markers [24]. The upper limb muscle strength, sit-and-reach, and one-minute sit-up tests are part of the validated EUROFIT test battery [25]. Endurance run is a local adaptation of the six-minute walk test, which was shown to have good reliability and validity for exercise tolerance and endurance [26]. The primary reason for the change from the walk to endurance run was to reduce the test’s ceiling effect and better differentiate the well-performing students. 

### 2.2. Statistical Analysis

All statistical analyses were performed using R Statistical Software version 3.6.3 (R Foundation for Statistical Computing, Vienna, Austria) (http://cran.us.r-project.org/, accessed on 5 September 2022). Potential outliers were examined and removed if the z-scores of the values were ≥3 standard deviations from the group mean by age and sex for each test.

BMI was calculated as the weight in kilograms (kg) divided by the square of height in meters (m^2^). Participants were categorized into underweight, normal-weight, overweight, and obese groups on the basis of their BMI according to the International Obesity Task Force (IOTF) age- and sex-specific standards [27]. International cutoff point passing through BMI 17 at age 18, which unifies with World Health Organization (WHO) standard of thinness, was used to define the underweight group. In contrast, the cutoff point passing through BMI 30 at age 18 was used to categorize the obese group. 

Descriptive statistics were used to examine participants’ characteristics, including age, sex, weight, height, and BMI category. The Kolmogorov–Smirnov test was used to determine the normality. A Mann–Whitney U test was used to compare the mean differences of continuous variables including in age, weight, height, BMI, and fitness scores. A Pearson’s chi-square test was used to examine the distributions of categorical variables, including school type and body status. A Kruskal–Wallis test was used to compare the group differences of fitness scores among BMI status. The Dunn test followed to test for pairwise comparisons. All tests were two-tailed, with *p* < 0.05 denoting statistical significance. The generalized additive model for location, scale, and shape (GAMLSS) [28], an extension of the LMS method developed by Cole and Green, was used to construct reference charts. The model consists of three components which are represented by four parameters: (1) location (µ), median; (2) scale (variability σ); and (3) shape (skewness ν and kurtosis τ). Distributions available include normal distribution (NO) with only location and scale; Box–Cox–Cole–Green (BCCG) distribution with location, scale, and skewness, where ν is equivalent to Box–Cox power λ; Box–Cox power–exponential (BCPE) distribution, which is an extension of BCCG distribution including kurtosis.

Power transformation (λ) was used to stretch or compress the age scale, i.e., $f\left(\lambda \right)=age^{\lambda },$ to obtain the optimal degree of freedom before modelling. The finesses of the models built on the basis of the above four distributions were compared using the Generalized Akaike Information Criterion (GAIC) and automatically selected by the *lms* function of the GAMLSS package (http://gamlss.org/, accessed on 5 September 2022) implemented in R version 3.6.3. The smallest GAIC with a penalty tree (GAIC(3)), which favours smoother curves, was chosen to provide an optimal fit for the model [29]. Q-Q plots of the normalized quantile residuals [30], Q statistics [31], and worm plots [32] were used to test the goodness of fit. Sex-specific sensitivity analysis was carried out by comparing the 5th, 10th, 20th, 30th, 40th, 50th, 60th, 70th, 80th, 90th, and 95th percentile curves for the fitness tests. P5, P10, P20, P30, P40, P50, P60, P70, P80, P90, and P95.

## 3. Results

### 3.1. Descriptive Characteristics

A total of 119,707 children and adolescents aged 6–17 years were recruited from September 2017 to June 2018. After removing outliers according to the aforementioned criteria, there were 119,693 (18% of sex- and age-matched population) participants included in this study. There were 89,438 (74.7%) participants from grade 1–6 (45,728 boys and 43,710 girls) and 30,255 (25.3%) participants from grade 7–12 (15,951 boys and 14,304 girls). The mean age of boys and girls was 10.94 years (3.02) and 10.85 years (2.96), respectively. 

Table 1 shows the descriptive characteristics of the subjects. The percentages of underweight and obese girls were 5.7% and 2.9%, respectively, and that of boys was 4.4% and 6.4%, respectively. Overall, there were significant sex differences in all fitness tests (all *p*-values < 0.01), with boys performing better than girls—handgrip strength: 26.67 (SD 9.59) kg vs. 25.13 (9.60) kg; push-up: 21 (12.44) reps vs. 13 (9.02) reps; sit-up: 28 (11.83) reps vs. 25 (10.24) reps; 6 min run–walk: 852.87 (172.35) m vs.811.9 (155.79) m; and 9 min run–walk: 1366.05 (289.76) m vs. 1239.72 (199.53) m—except for sit-and-reach 24.15 (7.77) cm vs. 29.48 (7.97) cm. 

#### BMI Status Comparison

Table 2 shows the differences in fitness test scores among BMI status by sex. There were significant differences in the test scores of all tests among groups. The mean test scores of the obese group were significantly lower than other groups (*p*-value < 0.01) except for handgrip strength, sit-and-reach in both boys and girls, and push-up in girls. For handgrip strength, the obese group performed the best (boys: 15.61(4.76) kg; girls: 14.71(5.65) kg). For sit-and-reach, there was no significant difference in the scores between the obese and underweight groups. The mean push-up test score of the obese girls was significantly lower than the overweight girls, and the mean push-up test score of the overweight girls was significantly lower than the underweight girls. The underweight group outperformed the obese group in push-ups, one-minute sit-ups, and endurance run tests, but not in handgrip strength.

### 3.2. The Fitted Models

Table 3 shows the summary of all the fitted models. The effective *df*(*μ*) of all models were greater than 2, indicating that the medians of fitness test scores did not have a linear relationship with age for either boys or girls. Similarly, most of the effective *df*(*ν*) was greater than 2, indicating that the skewness had no linear relationship with age.

### 3.3. Normative Values of Fitness Tests

Table 4a–f shows the normative values of all fitness tests tabulated as percentiles from 5 to 95. The smoothed centile curves (P_5_, P_10_, P_20_, P_30_, P_40_, P_50_, P_60_, P_70_, P_80_, P_90_, and P_95_) for the fitness scores by age and sex for children demonstrate that boys performed better in all fitness tests except the sit-and-reach test. The fitness scores increased steadily with age for the tests of muscle strength and endurance for both sexes. For boys, the sit-and-reach scores decreased at age 8 and increased at age 12. The development of the upper limb muscle power (handgrip strength and push-up) after 11 years old is faster in boys than in girls.

#### Centile Curves

Figure 1 shows the normative values of P_5_, P_10_, P_20_, P_30_, P_40_, P_50_, P_60_, P_70_, P_80_, P_90_, and P_95_ for boys and girls. There was a less pronounced positive age trend for push-ups, 1 min sit-ups, 6 min, and 9 min run–walk but a more positive age trend in handgrip strength in boys at the extreme percentiles.

## 4. Discussion

The present study established a new set of age and sex-specific references for Hong Kong Chinese children aged 6–17 years for a battery of physical fitness tests (upper limb muscle strength, abdominal muscle strength, flexibility, and cardiopulmonary endurance), which have utility for health and fitness screening. This reference, derived from the general population before the COVID-19 pandemic, allows for the accurate assessment of a child’s physical fitness level [33], identification of children with sporting success potential [34], and provision of guidance to those who are less fit to promote a healthier lifestyle. Based on the physical fitness references, the monitoring and training should be confined not only to the individual level but also to the population or group level to support, intervene, and improve those groups who are less fit physically. In addition, our findings demonstrated a nonlinear relationship between BMI status and physical fitness.

### 4.1. Classification of Fitness

Percentile charts are clinical tools used to identify the normal and outliers in various settings. To date, there have been no definite cutoff points for the tests used to determine physical fitness in pediatric populations. Children can be classified into five groups on the basis of their performance levels using the percentile chart: very low/poor (<P_20_), low (P_20_–P_40_), moderate (P_40_–P_60_), good (P_60_–P_80_), and very good (>P_80_). Although this classification is not criterion-referenced, the normative quintile-based framework was suggested by several multi-national European childhood studies [35,36,37]. Very low scores can indicate the need to develop proper fitness targets, follow up on long-term changes, encourage positive behaviours about health, and determine whether there are serious health problems. Percentiles are easier to understand and utilize in practice. Meanwhile, with the GAMLSS models, z-scores can be calculated and are more useful in research [38]. Most of the fitness tests used in today’s schools emphasize health-related fitness and are criterion-based rather than norm-based [34]. A reference is required to understand the genuine progress and improvement in specific areas of physical fitness. Monitoring physical fitness with a reference provides a wide range of health benefits [39]. On the basis of this information, children can receive advice and improvement strategies tailored to their fitness level. 

The centile curves are shifted upwards in endurance run tests, sit-ups, and push-ups, and are more obvious in boys. The association between physical fitness level and BMI status varies depending on the test used. Among four BMI statuses, the underweight group outperformed the obese group because their lower weight is advantageous for endurance running and sit-ups. Similar to previous studies, students who were overweight or obese performed worse on these tests than those who were normal weight [17,40]. According to the biological causes of obesity, excess weight and body fat make it harder for people to tolerate exercise and have lower aerobic capacities than children who are of normal weight [17]. By contrast, handgrip strength increased with BMI status, as reflected by BMI, which has been reported by previous studies [41,42]. This could be a result of obese children having higher fat-free mass and underweight children having lower fat-free mass. Meanwhile, a recent report showed that childhood handgrip strength was positively associated with BMI but negatively associated with body fat [43]. This highlights the importance that, apart from categorizing children’s weight status using BMI cutoffs, additional information on children’s body composition, such as percentage of body fat or fat-free mass, should also be considered. Obesity demonstrates a direct relation with handgrip strength, while other fitness tests demonstrate an inverse relation. This correlation between BMI status and physical fitness for promoting health has also been reported by Yi et al. [15]. Our results show that there is a nonlinear relationship between BMI and physical fitness, especially when the full range of BMI status, from underweight to obesity, is taken into account. In tests of muscular strength (handgrip, push-up, and sit-up) and cardiovascular fitness (endurance run), boys outperformed girls. In line with the literature [9,15,17], girls outperformed boys in the flexibility sit-and-reach test.

### 4.2. Health Education

Cale et al. suggested that, if integrated appropriately, fitness tests could contribute to the physical education curriculum and play a positive role in supporting healthy lifestyles in youths [44]. Mahar and Rowe also reported that individualized fitness tests and feedback to students about their fitness levels can promote goal setting and educate them about healthy lifestyles and fitness [34]. Previous reports also indicated the importance of teaching the students to self-assess their fitness levels [36,45]. Moreover, fitness tests should be part of the overall educational process, not an isolated part of the physical education curriculum [39,46,47]. As such, students should be taught about the relevance and benefits of each fitness component to their bodily functions. For example, students should be taught that the endurance run test performance is a measure of cardiopulmonary fitness and that cardiopulmonary fitness is associated with good health. Attainable, meaningful, and fair criterion-referenced standards can motivate even less physically fit students to accept physical challenges such as achieving higher fitness levels. Schools should play a central role in the provision and promotion of physical activity and physical fitness in children along with other healthy behaviours, as children spend a majority of their time in the school setting [48].

### 4.3. Policy

Sedentary lifestyles and the popularity of screen-based activities in children may also have an impact on childhood physical fitness [49,50,51]. In Hong Kong, only approximately 8% of school-aged children fulfill the recommended physical activity levels set by the World Health Organization in 2013 (i.e., at least 60 min of moderate- to vigorous-intensity physical activity) [52]. Given the strong link between childhood physical inactivity and long-term major diseases, such as metabolic syndrome, obesity, and cardiovascular problems, urgent attention is required. A recent report stated that appropriate policy could encourage healthy eating and physical active [53]. Several policies have been implemented by the government in Hong Kong to combat childhood obesity and physical inactivity. The Department of Health launched the EatSmart@school.hk Campaign in 2006 in order to promote healthy eating among Hong Kong schools, containing the “school policy on healthy eating”. The Healthy School Policy was established in 2008 and aims to help students reach a state of physical, mental, and social well-being with a focus on developing students’ healthy lifestyles, positive attitudes and values, practical life skills, and refusal skills to resist temptation. To encourage an active lifestyle in schoolchildren, the SPFAS was established in 1990 by the Hong Kong Child Health Foundation and the Education Bureau. After a long development of 32 years, the scheme is now serving more than 100,000 students annually. With the assistance of the aforementioned stakeholders, there has been significant progress in the improvement of health in Hong Kong schoolchildren.

However, a recent article in Hong Kong reported that that the average daily accumulation of moderate- or above-intensity physical activity time for primary and secondary school students was 15 min and 30 min, respectively, far short of the World Health Organization’s recommendation of 60 min [54]. Furthermore, time spent in front of electronic screens increased significantly due to the use of electronic products: primary school students’ screen usage increased from 2 h a day to 7 h a day before the pandemic, while middle school students’ screen usage increased from 8 to 9 h a day after the epidemic [54]. This might be due to the intermittent school closures since February 2020. School closure policy is one of the key infection control measures in Hong Kong during the pandemic, but it can impact children’s access to school meals and physical activity times, widening inequalities [55,56]. Changes in government policy appear to have a greater impact on children in terms of health. Therefore, the need for policies that support a healthy diet and regular physical activity in schools is highlighted by rising obesity rates and physical inactivity rates among schoolchildren, especially after the pandemic. The physical fitness reference established in the present study without the impact of the pandemic should be an important tool for the monitoring the trend of the physical fitness in Hong Kong children over time.

### 4.4. Strengths and Limitations

The present study has several strengths. First, the study had a relatively large sample size of more than 110,000 Chinese children aged 6–17, that is, 18% of the sex- and age-matched population. This age range represents the critical developmental period during which effective interventions are most likely to improve physical fitness levels [57]. Secondly, the references were developed using a well-refıned empirical methodology that included the use of GAMLSS models and standardized protocol of the fitness tests applied in this study. The use of GAMLSS models enabled the creation of standards that reflect sex differences and expected changes in growth and maturation. The normative curve smoothing method was used in this study to obtain the specific percentiles. Finally, the data were collected prior to the COVID-19 pandemic; it can serve as a baseline for assessing the impact of the COVID-19 pandemic on the physical fitness of Hong Kong schoolchildren.

However, a limitation of this study is that all our study participants were Hong Kong Chinese children living in an urban environment. Future collaborative studies involving Chinese children from rural and urban areas are needed to reflect the geographical variations of physical fitness among Chinese children. However, the defined norm values could be applied to children living in other urban areas of southern China. This cross-sectional study does not allow for the assessment of the effect of puberty on physical fitness. Therefore, longitudinal research should be performed to track the physical fitness level changes over time. Furthermore, the pubertal staging was not measured in this study. Controlling for this variable would help to minimize the variability among individuals of the same chronological age. In addition, weight not excluding fat weight was used in this study. Children might be classified into the obese group due to muscularity rather than fat. Therefore, further study using body fat percentage instead of BMI to define the obese group is needed. Moreover, without a standardized youth fitness testing battery, youth fitness changes cannot be monitored effectively. Physical fitness assessment tools vary between studies conducted in different regions; therefore, a direct comparison of childhood physical fitness from different regions becomes challenging [58]. By establishing a standardized test, countries would learn and borrow optimal test components from each other. While optional items can fit the diverse needs of schools, such inconsistency hinders the possible comparisons and improvement of fitness across countries.

## 5. Conclusions

This study established sex- and age-specific normative physical fitness values of Hong Kong Chinese children aged 6–17 years. Boys performed better in all measurements, except flexibility, than girls of the same age, and older children performed better than younger ones. These reference data will facilitate identifying children with low fitness, warranting guidance for positive health behaviours and setting up appropriate exercise goals. Our findings can inform future health education and physical fitness programmes in schools and epidemiology research on this topic. It also serves as a baseline for assessing the impact of the COVID-19 pandemic on Hong Kong students’ physical fitness in future studies.

## Figures and Tables

**Figure 1 ijerph-19-15346-f001:**
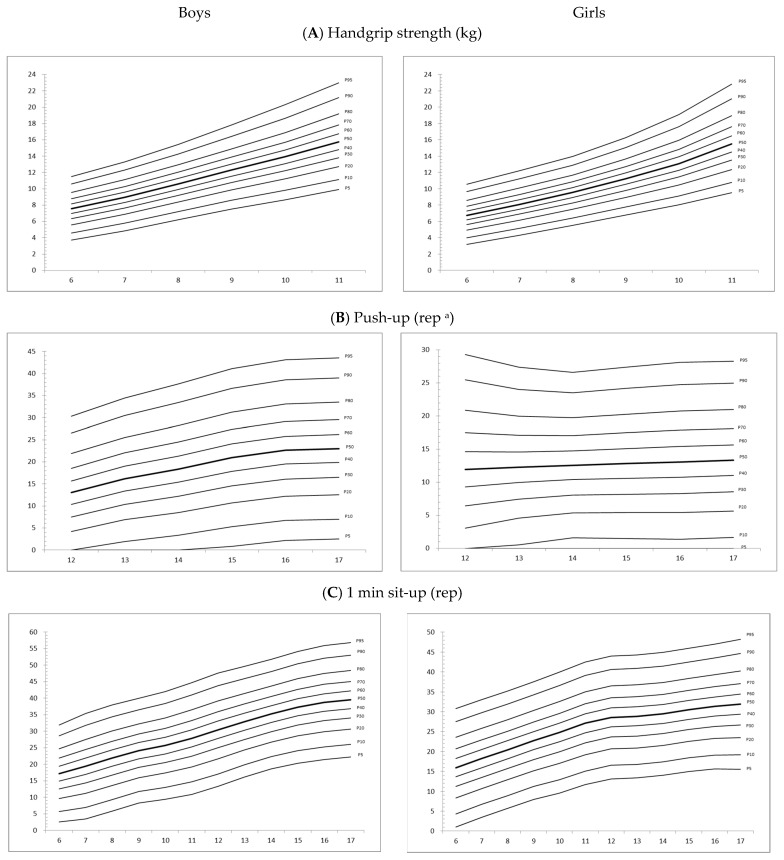
Normative value of handgrip strength, push-up, 1 min sit-up, sit-and-reach, 6 min and 9 min run–walk for boys and girls. ^a^ Modified knee-bend push-up was used for girls.

**Table 1 ijerph-19-15346-t001:** Subjects’ descriptive characteristics.

	All (n = 119,693)		Boys (n = 61,679)		Girls (n = 58,014)		
	n/Mean	%/Sd	n/Mean	%/Sd	n/Mean	%/Sd	*p*-Value ^
School type							
Primary	89,438	74.70%	45,728	74.10%	43,710	75.30%	<0.01 **
Secondary	30,255	25.30%	15,951	25.90%	14,304	24.70%	
Age (year)	10.89	2.99	10.94	3.02	10.85	2.96	<0.01 **
6	9366	7.80%	4802	7.80%	4564	7.90%	
7	10,902	9.10%	5651	9.20%	5251	9.10%	
8	14,724	12.30%	7513	12.20%	7211	12.40%	
9	16,700	14.00%	8383	13.60%	8317	14.30%	
10	16,975	14.20%	8684	14.10%	8291	14.30%	
11	16,535	13.80%	8450	13.70%	8085	13.90%	
12	7746	6.50%	3965	6.40%	3781	6.50%	
13	5423	4.50%	2837	4.60%	2586	4.50%	
14	5240	4.40%	2808	4.60%	2432	4.20%	
15	5488	4.60%	2899	4.70%	2589	4.50%	
16	5377	4.50%	2900	4.70%	2477	4.30%	
17	5217	4.40%	2787	4.50%	2430	4.20%	
Height (cm)	142.98	16.46	144.23	17.66	141.64	14.96	<0.01 **
Weight (kg)	39.31	19.14	39.96	23.4	36.54	12.94	<0.01 **
BMI (kg/m^2^)	18.06	6.95	18.46	9.06	17.64	3.47	<0.01 **
Body status #							
Underweight	5969	5.00%	2688	4.40%	3281	5.70%	<0.01 **
Normal	90,859	75.90%	44,404	72.00%	46,455	80.10%	
Overweight	17,257	14.40%	10,666	17.30%	6591	11.40%	
Obese	5608	4.70%	3921	6.40%	1687	2.90%	
Fitness scores							
Handgrip strength (kg)	12.98	4.78	13.36	4.76	12.59	4.76	<0.01 **
Push-up (Reps ^a^)	17.86	11.9	20.9	12.44	12.95	9.02	<0.01 **
1 min sit-up (Reps)	26.67	11.17	27.93	11.83	25.31	10.24	<0.01 **
Sit-and-reach (cm)	26.72	8.31	24.14	7.77	29.48	7.97	<0.01 **
6 min run–walk (m)	833.07	165.81	852.87	172.35	811.9	155.79	<0.01 **
9 min run–walk (m)	1302.08	256.11	1366.08	289.77	1239.72	199.53	<0.01 **

^ Mann–Whitney test was used for continuous variables and Pearson’s chi-square test was used for categorical variables. ** *p*-value < 0.01; # body status was accessed by IOTF reference; ^a^ Modified knee-bend push-up was used for girls.

**Table 2 ijerph-19-15346-t002:** Comparison of fitness test scores among BMI status groups.

Boys	Total (n = 61,679)		Underweight (n = 2688)		Normal (n = 44,404)		Overweight (n = 10,666)		Obese (n = 3921)			
	n/Mean	%/Sd	n/Mean	%/Sd	n/Mean	%/Sd	n/Mean	%/Sd	n/Mean	%/Sd	F	*p*-Value
Fitness scores												
Handgrip strength (kg)	13.36	4.76	11.26 _a_	3.80	12.85 _b_	4.54	15.13 _c_	4.90	15.61 _d_	4.76	836.13	<0.001
Push-up (Reps)	20.90	12.44	19.57 _a_	12.19	22.60 _b_	12.30	17.29 _a_	11.30	11.82 _c_	10.80	81.41	<0.001
1 min sit-up (Reps)	27.93	11.83	27.73 _a_	11.62	28.90 _b_	12.07	26.06 _c_	10.66	22.30 _d_	9.98	479.72	<0.001
Sit-and-reach (cm)	24.14	7.77	22.87 _a_	7.72	24.52 _b_	7.74	23.47 _c_	7.80	22.57 _a_	7.73	136.23	<0.001
6 min run–walk (m)	852.87	172.35	861.05 _a_	163.48	866.49 _a_	171.14	817.65 _b_	160.43	751.36 _c_	174.20	67.25	<0.001
9 min run–walk (m)	1366.08	289.77	1403.73 _a_	280.76	1416.83 _a_	284.25	1248.43 _b_	252.30	1122.56 _c_	229.07	712.84	<0.001
**Girls**	**Total (n = 58,014)**		**Underweight (n = 3281)**		**Normal (n = 46,455)**		**Overweight (n = 6591)**		**Obese (n = 1687)**			
	**n/Mean**	**%/Sd**	**n/Mean**	**%/Sd**	**n/Mean**	**%/Sd**	**n/Mean**	**%/Sd**	**n/Mean**	**%/Sd**	**F**	***p*-Value**
Fitness scores												
Handgrip strength (kg)	12.59	4.76	10.78 _a_	3.78	12.35 _b_	4.62	14.52 _c_	5.17	14.71 _d_	5.65	507.40	<0.001
Push-up (Reps ^a^)	12.95	9.02	13.12 _a,b_	9.59	13.27 _b_	9.00	11.12 _a,c_	8.70	9.27 _c_	7.90	6.89	<0.001
1 min sit-up (Reps)	25.31	10.24	24.73 _a_	10.08	25.76 _b_	10.34	23.57 _c_	9.40	20.88 _d_	9.25	195.89	<0.001
Sit-and-reach (cm)	29.48	7.97	27.95 _a_	7.53	29.70 _b_	7.99	29.01 _a_	7.93	28.22 _a_	7.74	72.11	<0.001
6 min run–walk (m)	811.90	155.79	823.37 _a_	149.86	818.29 _a_	155.44	780.19 _b_	149.01	727.70 _c_	160.20	29.14	<0.001
9 min run–walk (m)	1239.72	199.53	1260.56 _a_	194.41	1256.79 _a_	196.09	1148.52 _b_	183.72	1071.53 _c_	195.85	344.68	<0.001

Letter subscripts indicate significant differences between lettered groups using post hoc Dunn test. ^a^ Modified knee-bend push-up was used for girls.

**Table 3 ijerph-19-15346-t003:** GAMLSS models for fitness tests among 6 to 17-year-old Hong Kong children and adolescents.

Fitness Test	Distribution	Link	λ	*df*(*μ*)	*df*(*σ*)	*df*(*ν*)	*df*(*τ*)	*df*	Deviance	AIC	SBC
Boys											
Handgrip strength (kg)	BCT	log	1.50	7.14	6.14	3.34	4.67	21.28	208,931.31	208,960.52	209,143.46
Push-up (Reps)	NO	identity	1.50	4.39	2.65	-	-	7.04	26,549.29	26,560.44	26,603.60
1 min sit-up (Reps)	NO	identity	0.78	9.20	8.47	-	-	17.66	415,421.28	415,442.86	415,600.62
Sit-and-reach (cm)	NO	identity	1.30	8.89	5.65	-	-	14.53	387,289.77	387,308.72	387,438.65
6 min run–walk (m)	BCPE	log	1.50	7.51	4.41	2.00	4.50	18.42	75,799.70	75,789.74	75,912.54
9 min run–walk (m)	BCT	log	1.50	7.54	3.42	3.74	4.77	19.47	253,813.18	253,840.12	253,992.32
Girls											
Handgrip strength (kg)	BCT	log	1.50	8.34	5.99	3.94	3.04	21.32	205,631.59	205,661.01	205,844.34
Push-up (Reps ^a^)	NO	identity	0.60	4.04	4.25	-	-	8.29	15,811.04	15,823.62	15,870.78
1 min sit-up (Reps)	NO	identity	1.33	9.83	6.26	-	-	16.09	393,160.19	393,180.17	393,323.30
Sit-and-reach (cm)	BCPE	log	0.35	10.36	8.50	6.98	3.44	29.28	373,870.47	373,910.68	374,171.48
6 min run–walk (m)	BCPE	log	1.50	5.00	5.00	5.00	5.00	20.00	72,002.41	72,030.41	72,163.09
9 min run–walk (m)	BCPE	log	5.85	6.39	4.40	6.57	2.03	19.38	260,394.30	260,421.55	260,574.30

^a^ Modified knee-bend push-up was used for girls.

**Table 4 ijerph-19-15346-t004:** Handgrip (kg) centile values by sex and age in 6 to 11-year-old Hong Kong children and adolescents. (**a**). Handgrip (kg) centile values by sex and age in 6 to 11-year-old Hong Kong Children and Adolescents. (**b**). Push-up (Reps ^a^) centile values by sex and age in 12 to 17-year-old Hong Kong children and adolescents. (**c**). Sit-and-reach (cm) centile values by sex and age in 6 to 17-year-old Hong Kong children and adolescents. (**d**). 1 min sit-up (Reps) centile values by sex and age in 6 to 17-year-old Hong Kong children and adolescents. (**e**). 6 min run–walk (m) centile values by sex and age in 6 to 8-year-old Hong Kong children and adolescents. (**f**). 9 min run–walk (m) centile values by sex and age in 9 to 17-year-old Hong Kong children and adolescents.

**(a)**											
**Age (year)**	**P_5_**	**P_10_**	**P_20_**	**P_30_**	**P_40_**	**P_50_**	**P_60_**	**P_70_**	**P_80_**	**P_90_**	**P_95_**
Boys											
6	3.71	4.56	5.59	6.33	6.96	7.55	8.15	8.79	9.54	10.61	11.5
7	4.87	5.78	6.87	7.66	8.33	8.97	9.61	10.3	11.13	12.31	13.32
8	6.22	7.19	8.36	9.2	9.92	10.6	11.28	12.04	12.95	14.27	15.44
9	7.52	8.59	9.86	10.77	11.56	12.3	13.05	13.89	14.91	16.44	17.85
10	8.64	9.81	11.22	12.23	13.1	13.93	14.78	15.72	16.89	18.67	20.32
11	9.91	11.16	12.7	13.83	14.82	15.77	16.75	17.84	19.17	21.17	22.98
Girls											
6	3.17	3.98	4.93	5.61	6.19	6.74	7.28	7.88	8.6	9.64	10.56
7	4.29	5.15	6.17	6.9	7.51	8.1	8.68	9.32	10.1	11.22	12.22
8	5.53	6.44	7.51	8.28	8.93	9.55	10.17	10.86	11.68	12.89	13.96
9	6.77	7.77	8.96	9.81	10.54	11.23	11.94	12.71	13.65	15.04	16.28
10	8.02	9.13	10.47	11.44	12.28	13.09	13.91	14.82	15.94	17.6	19.09
11	9.52	10.79	12.36	13.52	14.53	15.5	16.5	17.6	18.97	21.01	22.85
**(b)**											
**Age (year)**	**P_5_**	**P_10_**	**P_20_**	**P_30_**	**P_40_**	**P_50_**	**P_60_**	**P_70_**	**P_80_**	**P_90_**	**P_95_**
Boys											
12	0	0	4	7	10	13	15	18	21	26	30
13	0	1	6	10	13	16	19	22	25	30	34
14	0	3	8	12	15	18	21	24	28	33	37
15	0	5	10	14	17	20	24	27	31	36	41
16	2	6	12	16	19	22	25	29	33	38	43
17	2	7	12	16	19	23	26	29	33	39	43
Girls											
12	0	0	3	6	9	11	14	17	20	25	29
13	0	0	4	7	9	12	14	17	20	24	27
14	0	1	5	8	10	12	14	17	19	23	26
15	0	1	5	8	10	12	15	17	20	24	27
16	0	1	5	8	10	13	15	17	20	24	28
17	0	1	5	8	11	13	15	18	21	25	28
**(c)**											
**Age (year)**	**P_5_**	**P_10_**	**P_20_**	**P_30_**	**P_40_**	**P_50_**	**P_60_**	**P_70_**	**P_80_**	**P_90_**	**P_95_**
Boys											
6	14.23	16.39	19	20.88	22.49	23.99	25.5	27.11	28.99	31.6	33.75
7	14.15	16.41	19.13	21.1	22.78	24.35	25.92	27.6	29.56	32.29	34.54
8	13.72	16.08	18.94	21	22.76	24.41	26.06	27.82	29.88	32.74	35.1
9	12.79	15.28	18.29	20.46	22.32	24.05	25.79	27.64	29.81	32.82	35.31
10	11.77	14.37	17.53	19.81	21.75	23.57	25.39	27.34	29.61	32.77	35.38
11	10.77	13.49	16.79	19.17	21.2	23.09	24.99	27.02	29.4	32.7	35.42
12	10.1	12.96	16.42	18.91	21.04	23.03	25.02	27.16	29.65	33.11	35.96
13	9.97	12.99	16.64	19.28	21.53	23.63	25.74	27.99	30.63	34.28	37.3
14	9.73	12.95	16.85	19.66	22.06	24.3	26.55	28.95	31.76	35.65	38.87
15	9.75	13.16	17.29	20.27	22.82	25.2	27.58	30.13	33.11	37.24	40.65
16	10.05	13.6	17.9	21	23.65	26.12	28.6	31.25	34.35	38.65	42.2
17	10.28	13.92	18.32	21.5	24.21	26.75	29.28	31.99	35.17	39.57	43.21
Girls											
6	15.98	18.76	21.82	23.86	25.51	26.98	28.4	29.87	31.52	33.71	35.45
7	16.63	19.29	22.32	24.38	26.09	27.64	29.14	30.69	32.45	34.79	36.65
8	16.84	19.49	22.55	24.68	26.46	28.09	29.69	31.34	33.21	35.7	37.67
9	16.23	18.98	22.2	24.47	26.38	28.15	29.88	31.67	33.69	36.36	38.48
10	15.57	18.48	21.92	24.37	26.45	28.39	30.28	32.24	34.44	37.34	39.62
11	15.39	18.47	22.15	24.81	27.08	29.21	31.29	33.43	35.83	38.97	41.43
12	15.55	18.81	22.75	25.6	28.06	30.35	32.6	34.91	37.47	40.79	43.37
13	16.27	19.69	23.78	26.74	29.27	31.63	33.93	36.27	38.82	42.1	44.6
14	17.3	20.87	25.1	28.13	30.71	33.1	35.4	37.71	40.2	43.34	45.7
15	18.04	21.77	26.12	29.21	31.83	34.24	36.55	38.83	41.27	44.29	46.54
16	18.43	22.26	26.72	29.86	32.53	34.96	37.28	39.56	41.97	44.92	47.09
17	18.7	22.55	27.03	30.19	32.86	35.3	37.6	39.86	42.23	45.1	47.19
**(d)**											
**Age (year)**	**P_5_**	**P_10_**	**P_20_**	**P_30_**	**P_40_**	**P_50_**	**P_60_**	**P_70_**	**P_80_**	**P_90_**	**P_95_**
Boys											
6	3	6	10	13	15	17	19	22	25	29	32
7	3	7	11	14	17	19	22	24	28	32	35
8	6	9	14	17	19	22	24	27	30	34	38
9	8	12	16	19	22	24	27	29	32	36	40
10	9	13	17	20	23	26	28	31	34	38	42
11	11	15	19	22	25	28	30	33	36	41	45
12	13	17	22	25	28	30	33	36	39	44	48
13	16	20	24	28	30	33	35	38	41	46	50
14	19	22	27	30	33	35	38	41	44	48	52
15	20	24	29	32	35	37	40	43	46	50	54
16	21	25	30	33	36	39	41	44	48	52	56
17	22	26	31	34	37	40	42	45	48	53	57
Girls											
6	1	4	8	11	14	16	18	21	24	28	31
7	3	7	11	14	16	18	21	23	26	30	33
8	6	9	13	16	18	20	23	25	28	32	35
9	8	11	15	18	20	23	25	27	30	34	38
10	10	13	17	20	22	25	27	30	33	37	40
11	12	15	19	22	25	27	29	32	35	39	43
12	13	17	21	24	26	29	31	34	36	41	44
13	13	17	21	24	26	29	31	34	37	41	44
14	14	17	22	25	27	29	32	34	37	42	45
15	15	18	23	26	28	30	33	35	38	43	46
16	16	19	23	26	29	31	34	36	39	44	47
17	16	19	24	27	29	32	34	37	40	45	48
**(e)**											
**Age (year)**	**P_5_**	**P_10_**	**P_20_**	**P_30_**	**P_40_**	**P_50_**	**P_60_**	**P_70_**	**P_80_**	**P_90_**	**P_95_**
Boys											
6	560.64	624.84	694.71	739.43	773.83	803.03	832.32	867.09	912.69	984.87	1052.28
7	566.79	630.43	702.24	749.99	787.98	821.3	854.82	893.55	943.04	1019.18	1088.52
8	577.52	636.99	707.56	757.1	798.39	836.21	874.41	916.99	969.33	1046.42	1113.85
Girls											
6	527.59	582.15	645.66	690.13	727.42	761.68	795.1	829.65	868.56	920.06	960.73
7	551.43	609.55	675.2	719.03	754.1	785.06	816.08	851.37	895.73	962.66	1022.44
8	578.56	636.8	702.76	747.02	782.59	814.15	845.72	881.3	925.6	991.68	1050.05
**(f)**											
**Age (year)**	**P_5_**	**P_10_**	**P_20_**	**P_30_**	**P_40_**	**P_50_**	**P_60_**	**P_70_**	**P_80_**	**P_90_**	**P_95_**
Boys											
9	786.51	878.98	986.9	1061.64	1123.37	1179.49	1235.69	1297.65	1372.92	1482.08	1576.07
10	821.13	909.77	1014.81	1088.65	1150.36	1207.04	1263.96	1326.47	1402.02	1510.98	1604.35
11	866.82	953.16	1057.02	1131.15	1193.87	1252.08	1310.68	1374.69	1451.54	1561.51	1655.09
12	920.56	1006.93	1111.95	1187.79	1252.59	1313.19	1374.24	1440.5	1519.41	1631.19	1725.43
13	977.96	1066.86	1175.3	1253.94	1321.38	1384.64	1448.24	1516.81	1597.79	1711.33	1806.12
14	1031.27	1125.47	1239.58	1321.87	1392.19	1457.91	1523.68	1594.15	1676.75	1791.53	1886.51
15	1077.08	1179.09	1300.7	1387.18	1460.32	1528.11	1595.5	1667.34	1751.14	1866.89	1962.07
16	1104.69	1213.99	1341.9	1431.39	1506.17	1574.83	1642.67	1714.75	1798.58	1913.98	2008.53
17	1109.53	1223.35	1354.51	1445.06	1520.03	1588.35	1655.51	1726.61	1809.03	1922.01	2014.2
Girls											
9	757.06	853.98	960.08	1030	1085.92	1135.51	1183.85	1235.5	1296.14	1380.47	1450.12
10	843.67	920.26	1009.33	1070.84	1121.56	1167.6	1213.62	1264.23	1325.51	1414.03	1489.97
11	897.63	965.63	1046.59	1103.61	1151.23	1194.95	1239.24	1288.8	1350.04	1440.81	1520.75
12	927.36	997.81	1080.57	1138.1	1185.67	1228.96	1272.61	1321.42	1381.65	1470.73	1548.95
13	957.5	1030.68	1115.33	1173.35	1220.8	1263.58	1306.5	1354.42	1413.48	1500.6	1576.84
14	975.76	1051.91	1138.38	1196.66	1243.72	1285.68	1327.55	1374.25	1431.72	1516.27	1590.01
15	980.04	1061.77	1152.08	1211.51	1258.68	1300.14	1341.16	1386.8	1442.8	1524.76	1595.78
16	979.47	1067.53	1162.2	1223.05	1270.54	1311.71	1352.16	1397.13	1452.22	1532.63	1602.02
17	984.86	1074.71	1170.26	1231.06	1278.13	1318.63	1358.39	1402.82	1457.53	1537.84	1607.48

(**a**,**c**–**e**) Note: Ages shown represent as completed age (e.g., 6 = 6.00–6.99). (**b**) Note: Ages shown represent as completed age (e.g., 12 = 12.00–12.99); ^a^ Modified knee-bend push-up was used for girls. (**f**) Note: Ages shown represent as completed age (e.g., 9 = 9.00–9.99).

## Data Availability

All data that support the findings of this study are available from the corresponding author upon reasonable request.

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
