# Peer review of "Age- and Sex-Specific Physical Fitness Reference and Association with Body Mass Index in Hong Kong Chinese Schoolchildren"

_ijerph, 2022, doi:10.3390/ijerph192215346_

Round 1

Reviewer 1 Report

It is an interesting and important study. The manuscript is well-written and easy to read. The aim of the study was to establish the age and sex-specific fitness references using a representative sample of children and adolescents aged 6-17 years and to determine the associations with body mass index. The study adds important new knowledge for public health.

There are my comments for the manuscript:

The study is cross-sectional and causal language should be avoided in the study‘s aim. The associations between BMI and fitness performance were explored in this study. Obviously, higher BMI prevents children from better performance in the majority of fitness tests, however, it might be that a sedentary lifestyle, lower physical activity and poor fitness (as an outcome) impact the development of overweight and/or obesity in children. Therefore, I suggest revising the aim of the study (line 68. p. 3) since the association might be reciprocal.

The authors state that „increasing childhood obesity prevalence has shifted the classification of healthy fitness, with „underfit“ as normal“ (lines 60-61, p. 3).  The development of obesity might have a biopsychosocial background, therefore, it is important for public health not to focus on obesity as the main cause of poor fitness. The observed negative shift in the classification of healthy fitness might also be influenced by a sedentary lifestyle and the popularity of screen-based activities in children and youth. Thus, I recommend revising this sentence and expanding the discussion on complex reasons for fitness decline in children and adolescents not focusing only on obesity.

Since the classification of healthy fitness declines to shift from „underfit“to „normal“, the question is why authors use the sample of 2017-2018, but not from databases of earlier fitness tests (if they are available)? Is it the first attempt to create fitness standards for Hong Kong‘s youth population? A deeper discussion on this topic in the Introduction and Discussion sections is necessary.

Author Response

Thank you so much for taking the time to review this manuscript and making insightful comments!

The point-to-point responses to your comments are as follows.

Reviewer I

It is an interesting and important study. The manuscript is well-written and easy to read. The aim of the study was to establish the age and sex-specific fitness references using a representative sample of children and adolescents aged 6-17 years and to determine the associations with body mass index. The study adds important new knowledge for public health.

There are my comments for the manuscript:

The study is cross-sectional and causal language should be avoided in the study‘s aim. The associations between BMI and fitness performance were explored in this study. Obviously, higher BMI prevents children from better performance in the majority of fitness tests, however, it might be that a sedentary lifestyle, lower physical activity and poor fitness (as an outcome) impact the development of overweight and/or obesity in children. Therefore, I suggest revising the aim of the study (line 68. p. 3) since the association might be reciprocal.

Response: Thank you for the comment. We changed the “determine the impact of BMI status on physical fitness “ to “determine the associations with body mass index” accordingly.

The authors state that „increasing childhood obesity prevalence has shifted the classification of healthy fitness, with „underfit“ as normal“ (lines 60-61, p. 3).  The development of obesity might have a biopsychosocial background, therefore, it is important for public health not to focus on obesity as the main cause of poor fitness. The observed negative shift in the classification of healthy fitness might also be influenced by a sedentary lifestyle and the popularity of screen-based activities in children and youth. Thus, I recommend revising this sentence and expanding the discussion on complex reasons for fitness decline in children and adolescents not focusing only on obesity.

Response: “Increasing childhood obesity prevalence has shifted the classification of healthy fitness, with ‘underfit’ as normal, and this process of normalization is not benign.” has been deleted and further discussion on complex reasons for fitness decline in children and adolescents not focusing only on obesity has been added in the discussion.

Since the classification of healthy fitness declines to shift from “underfit“ to  “normal“, the question is why authors use the sample of 2017-2018, but not from databases of earlier fitness tests (if they are available)? Is it the first attempt to create fitness standards for Hong Kong‘s youth population? A deeper discussion on this topic in the Introduction and Discussion sections is necessary.

Response: Beginning in 2013, the School Physical Fitness Award Scheme (SPFAS) established an electronic platform to collect anthropometric and fitness data from participating students. The current study retrieved and analyzed SPFAS data on fitness and anthropometric measurements recorded by grade 1 to 12 students from 2017 to 2018 under the SPFAS because it is the most recent complete dataset and should be a representative sample of 119,707 children and adolescents aged 6-17 years. It is the first population-based study of fitness in Hong Kong children. That is also addressed in the introduction and discussion. By 2005, the overall prevalence of childhood overweight was 17.8% in Hong Kong which represents an approximate increase of 1.5-fold over 10 years. Since 2010, the overweight and obesity rate of Hong Kong students decreased gradually and reach to a plateau in 2017.It seems has a raising trend after the pandemic but In view of serious service disruption due to COVID-19. Direct comparison of the data in 2019/20 with that of previous years has to be done with caution. Therefore, we don’t include the information in the manuscript.

Reference:

Lau P.W., Yip T.C. Childhood obesity in Hong Kong: a developmental perspective and review, 1986–2005. J Exerc Sci Fit. 2006;4(2):67–84.

Department of Health of Hong Kong. 2019. Overweight and Obesity. Available at: https://www.chp.gov.hk/en/statistics/data/10/757/5513.html

Reviewer 2 Report

·     Title: suggest to revise- “Association between physical fitness and body mass index in Hong Kong Chinese schoolchildren”.

·         The abstract should be rewritten and included the following: (1) a background to make a reader understand the reasons of conducting this study; (2) the type of study design (i.e. cross-sectional, longitudinal); (3) the place/region (i.e. Northern HK) where the study was conducted; (4) the term “reference” should be clearly defined”; (5) the type of statistical analysis used; and (6) P-values or statements about the significance of results.

·         Introduction: This section is very short. Few cross-sectional/longitudinal studies were used to support the aim of the study. The novelty and/or significance of this study should be described. What does the study add to the literature? Why would physical fitness be associated with BMI? What could possibly be the causal mechanism? In general, the paper readability could be improved by rewriting this section aimed to increase its length.

·         The methods are introduced only briefly but in a few cases without enough clarity. I feel a further description of the methodology that authors used to analyze the data would be beneficial. What was the type of study design? What were the inclusion and exclusion criteria? How children were recruited? How did the authors deal with missing values? Data collection procedure should be described in much more details.

·         I recommend including more details of the analysis in terms of variables used (i.e. continuous, categorical), as the statistical analysis section should be clear enough. Authors should clarify each statistical test used and the rationale for using those tests. Did the authors check for normality distribution?

·         Clear rational why the data were analyzed according to gender?

·         It is unclear from Tables 1,3 & 4 what test was used?

·         The results within the text should be reported in much more details.

·         The discussion is way too short and in some areas difficult to read as it is unclear what was found in the study and other studies that authors compared with.

·         It is good that authors include a “health education” section. However, it should be expanded to include policy recommendations to increase physical activity and reduce obesity among Hong Kong Chinese schoolchildren. Authors should discuss policies with anticipated outcomes for obesity problem, with some reflection on Chinese government. I would suggest referring this these articles (i.e., Children (Basel). 2018 Jan 29;5(2):18.doi 10.3390/children5020018; Int J Environ Res Public Health. 2020, 17(22):8405).

·         Moderate English language change required.

Author Response

Thank you so much for taking the time to review this manuscript and making insightful comments!

The point-to-point responses to your comments are as follows.

Reviewer II

Title: suggest to revise- “Association between physical fitness and body mass index in Hong Kong Chinese schoolchildren”.

Response: As we aim to establish a reference for Hong Kong children’s fitness, we have changed the title as “Age and sex-specific physical fitness reference and association with body mass index in Hong Kong Chinese schoolchildren.”

  • The abstract should be rewritten and included the following: (1) a background to make a reader understand the reasons of conducting this study; (2) the type of study design (i.e. cross-sectional, longitudinal); (3) the place/region (i.e. Northern HK) where the study was conducted; (4) the term “reference” should be clearly defined”; (5) the type of statistical analysis used; and (6) P-values or statements about the significance of results.

Response: The abstract has been revised accordingly.

  • Introduction: This section is very short. Few cross-sectional/longitudinal studies were used to support the aim of the study. The novelty and/or significance of this study should be described. What does the study add to the literature? Why would physical fitness be associated with BMI? What could possibly be the causal mechanism? In general, the paper readability could be improved by rewriting this section aimed to increase its length.

Response: The introduction has been rewritten accordingly and it has been divided into 3 paragraphs.

  • The methods are introduced only briefly but in a few cases without enough clarity. I feel a further description of the methodology that authors used to analyze the data would be beneficial. What was the type of study design? What were the inclusion and exclusion criteria? How children were recruited? How did the authors deal with missing values? Data collection procedure should be described in much more details.

Response: Present study is a cross-sectional study using existing anonymous records from a territory scheme. Only Chinese students aged between 6 and 17 were included. Students without anthropometric data or complete data of the fitness tests were excluded. Potential outliers were examined and removed if the z-scores of the values were ≥ 3 standard deviations from the group mean by age and sex for each test. Details have been added.

  • I recommend including more details of the analysis in terms of variables used (i.e. continuous, categorical), as the statistical analysis section should be clear enough. Authors should clarify each statistical test used and the rationale for using those tests. Did the authors check for normality distribution?

Response: Kolmogorov–Smirnov test was used to determine the normality. Details have been added.

  • Clear rational why the data were analyzed according to gender?

Response: According to fitness references constructed by other previous studies, there is significant difference between sex groups on fitness outcomes and this is also reassured in our study.

  • It is unclear from Tables 1,3 & 4 what test was used?

Response: Tests detailed was added in Table 1. Table 3 and 4 shows the model details of GAMLSS models and no comparison is performed.

  • The results within the text should be reported in much more details.

Response: Details have been added.

  • The discussion is way too short and in some areas difficult to read as it is unclear what was found in the study and other studies that authors compared with.

Response: The discussion has been revised.

  • It is good that authors include a “health education” section. However, it should be expanded to include policy recommendations to increase physical activity and reduce obesity among Hong Kong Chinese schoolchildren. Authors should discuss policies with anticipated outcomes for obesity problem, with some reflection on Chinese government. I would suggest referring this these articles (i.e., Children (Basel). 2018 Jan 29;5(2):18.doi 10.3390/children5020018; Int J Environ Res Public Health. 2020, 17(22):8405).

Response: A section of “Policy” has been added in the discussion and thank you for the suggestion of the reference.

  • Moderate English language change required.

Response: A native English speaker helps in the editing.

Round 2

Reviewer 2 Report

No further comments.